# Astaxanthin Induces Apoptosis in MCF-7 Cells through a p53-Dependent Pathway

**DOI:** 10.3390/ijms25137111

**Published:** 2024-06-28

**Authors:** Koanhoi Kim, Hyok-rae Cho, Yonghae Son

**Affiliations:** 1Department of Pharmacology, School of Medicine, Pusan National University, Busan 43241, Republic of Korea; koanhoi@pusan.ac.kr; 2Department of Neurosurgery, College of Medicine, Kosin University, Busan 49267, Republic of Korea

**Keywords:** apoptosis, astaxanthin, breast cancer, MCF-7, p53

## Abstract

Astaxanthin (3,3′-dihydroxy-β,β-carotene-4,4′-dione; AXT) is a xanthophyll β-carotenoid found in microalgae, seafood, fungi, complex plants, flamingos, and quail. It is well known that AXT plays a role as a drug with antioxidant and antitumor properties. Furthermore, several studies have reported that the reagent shows anti-inflammatory and neuroprotective effects. Recently, it was found that AXT acts as a peroxisome proliferator-activated receptor γ (PPARγ) modulator. To investigate the effect of AXT on MCF-7 cells (a human breast cancer cell line), the cells were treated with various concentrations of AXT. The treatment induced the decrease in cell number in a dose-dependent manner. Additionally, the Annexin V-positive cells were increased by the AXT treatment. These results indicated that apoptosis was induced in the tumor cells through the treatment of AXT. To elucidate the connection between apoptosis and p53, the levels of p53 and p21 proteins were assessed. Consequently, it was observed that the expression of p53 and p21 increased proportionally to the concentration of the AXT treatment. These findings suggest that the apoptosis of MCF-7 cells induced by AXT operates through a p53-dependent pathway, implying that AXT could potentially have a beneficial role in future breast cancer treatments. Thus, our results will provide a direction for future cancer challenges.

## 1. Introduction

Astaxanthin (3,3′-dihydroxy-β,β-carotene-4,4′-dione; AXT) is a xanthophyll carotenoid found in microalgae, fungi, higher plants, seafood, flamingos, and quail [1]. Studies have demonstrated its anti-inflammatory effects on LPS-induced uveitis and laser-induced choroidal neovascularization [2,3]. Furthermore, ongoing research explores its medical potential due to its antioxidative and antitumor properties [4,5]. A recent investigation indicated its involvement in insulin signaling in L6 cells [6]. According to this study, AXT may enhance glucose metabolism under insulin-resistant conditions, suggesting a potential positive role in insulin action. However, the precise mechanisms underlying AXT’s antitumor and antioxidative effects, including its receptors, target genes, and pathways, remain inadequately explored.

P53 is a well-established tumor suppressor that regulates the cell cycle by inducing G1 phase cell cycle arrest [7]. Phosphorylated p53 acts as a transcription factor, promoting the transcription of p21, which, in turn, induces cell cycle arrest. Activation of the p53/p21 pathway ultimately leads to cell apoptosis through protein upregulation [8,9]. Moreover, p53 is involved in DNA repair mechanisms [10,11,12]. Dysregulation, dysfunctionality, inactivation, mutation, or silencing of p53 are observed in various pathological conditions such as cancer, neurodegeneration, ischemia, cholestasis, or atherosclerosis [13]. 

In this study, we demonstrate that AXT inhibits breast cancer cell growth by inducing p53-mediated cell cycle arrest and programmed cell death.

## 2. Results

### 2.1. Roles of AXT in MCF-7 Viability and Apoptosis

To determine the IC50 of AXT, we performed treatment with different concentrations (5, 10, 25, 50, 75, 100, 125, and 150 μg/mL) on the MCF-7 cell line and assessed the cell viability. The results indicated that the concentration at which cell viability was 50% was 89.742 μg/mL (IC50 = 89.742 μg/mL). For subsequent experiments, we selected a concentration of 50 μg/mL, where cell viability was approximately 70% (Appendix A).

To examine the influence of AXT on MCF-7 cells, cell viability was assessed. MCF-7 cells were exposed to AXT at concentrations of 10, 50, and 100 μg/mL for the indicated times. Subsequently, cell viability was determined using PrestoBlue™ Cell Reagents, in accordance with the manufacturer’s guidelines. The outcomes demonstrated a dose- and time-dependent decline in cell viability in the presence of AXT (Figure 1A). Treatment with 50 μg/mL of AXT led to a viability of the tumor cell of approximately 30% after 48 h. Notably, strong toxicity was observed with 100 μg/mL of AXT for 48 h. These observations emphasize the significant impact of AXT on MCF-7 cell viability.

Next, in order to ascertain the role of AXT in p53-dependent cellular apoptosis, we conducted an analysis of the cell death. MCF-7 cells were subjected to a 48 h incubation period with 50 μg/mL of AXT, followed by examination using the MUSE^TM^ Cell Analyzer as outlined in the Materials and Methods section. While the baseline apoptosis rate in untreated MCF-7 cells was measured at 17.18%, this rate escalated to 26.48% following exposure to AXT (Figure 1B). Specifically, the late apoptosis rate exhibited a more pronounced increase upon AXT treatment. These findings strongly suggest that AXT induces the programmed cell death in tumor cells.

### 2.2. Roles of AXT in Cell Cycle and the Expression of Tumor Suppressor Proteins

We elucidated the role of AXT in the cell cycle. Following treatment with AXT, cells were harvested and analyzed using the MUSE^TM^ Cell Analyzer. The administration of AXT resulted in a significant increase in the G2/M phase rate, rising from 24.2% to 30.9% in MCF-7 cells (Figure 2A). This finding unequivocally demonstrates that AXT induces arrest in the late G2/M phase in MCF-7 cells.

We investigated whether AXT was implicated in the expression of p53, a tumor suppressor protein. MCF-7 cells were exposed to 50 μg/mL of AXT for 12, 24, and 48 h, after which total proteins extracted from the treated cells were subjected to Western blot analysis. The results revealed a significant upregulation of p53 expression after 48 h of AXT treatment (Figure 2B). Additionally, the expression of p21, a downstream target of p53, was concomitantly elevated following the increase in p53 levels. These findings suggest that AXT is involved in modulating the expression of p53.

### 2.3. Effects of AXT on Caspase-3 Activity

To investigate the involvement of AXT in p53-mediated apoptosis, we measured caspase-3 activity, a key protein in cell apoptosis. MCF-7 cells were treated with 50 μg/mL of AXT for the indicated durations, and protein activity was measured according to the kit’s instructions. The results indicated that caspase-3 activity increased following AXT treatment, reaching a maximum at 48 h post treatment (Figure 3A). We also examined whether caspases influenced AXT-induced apoptosis. Treatment with z-VAD-fmk (a pan-caspase inhibitor) and z-DEVD-fmk (a caspase-3/7 inhibitor) reversed the decrease in viability induced by AXT (Figure 3B). These findings suggest that AXT-induced apoptosis is mediated by caspase cascades.

## 3. Materials and Methods

### 3.1. Cell and Reagents

MCF-7, a human breast cancer cell line, was procured from American Type Cell Culture (ATCC; Manassas, VA, USA) and cultured in Dulbecco’s modified Eagle medium (DMEM) (HyClone Laboratories, Inc., Logan, UT, USA) supplemented with 10% fetal bovine serum (FBS) (Gibco Life Technologies Co., Grand Island, NY, USA). Primary antibodies against p53 and β-actin were obtained from Santa Cruz (Delaware Avenue, CA, USA), while the antibody against p21 was sourced from BD Biosciences (San Jose, CA, USA). AXT was acquired from Sigma (St. Louis, MO, USA) and dissolved in dimethyl sulfoxide (DMSO). z-VAD-fmk and z-DEVD-fmk were purchased from Cell Signaling Technology, Inc. (Danvers, MA, USA).

### 3.2. Western Blot

For the preparation of whole-cell lysates, MCF-7 cells were harvested 48 h post treatment with AXT. Subsequently, cellular lysis was accomplished utilizing PRO-PREP^TM^ (iNtRON Biotechnology, Seoul, Republic of Korea). The protein content of each lysate was quantified by employing the bicinchoninic acid method (BCA, Pierce, Rockford, IL, USA). The lysates were then subjected to SDS-PAGE and transferred onto nitrocellulose membranes (Whatman GmbH, Dassel, Germany). Following an hour-long incubation in 5% skim milk in 0.1% Tween 20/TBS to obstruct nonspecific binding sites, the membranes were probed with specified primary antibodies overnight at 4 °C. Subsequent to thrice washing with 0.1% Tween 20/TBS for 15 min each, the membranes were exposed to HRP-conjugated secondary antibodies for one hour at room temperature. Following another round of thrice washing with the washing buffer for 15 min each, protein bands were visualized utilizing the enhanced chemiluminescence (ECL) Western blotting detection system (Amersham Pharmacia Biotech, Piscataway, NJ, USA). β-Actin for the control group was detected using the same method as other proteins, involving the addition of an identical amount of measured protein to a gel of the same concentration.

### 3.3. Cell Viability Assay

To assess the viability of cells following AXT treatment, 5 × 10^3^ cells were seeded into a 96-well plate. These cells were then exposed to varying concentrations of AXT in growth media for 12, 24, and 48 h. Following a 48 h incubation period, cell viability was evaluated using PrestoBlue^TM^ cell viability reagents (Invitrogen, Carlsbad, CA, USA). In brief, reagents comprising 10% of the total media volume were introduced into the culture medium. After a 30 min incubation period, absorbance at 540 nm wavelength was measured using a spectrophotometer.

### 3.4. Analysis of Caspase-3 Activity

To determine the activity of caspase-3 on the AXT-exposed MCF-7, it was investigated with Caspase-3 Activity Assay Kit (Cell Signaling Technology). The experiment was performed according to the user manual of the kit.

### 3.5. Cell Cycle Analysis

The investigation of cell cycle dynamics was conducted utilizing the MUSE^TM^ Cell Analyzer system. Cells were cultured for a duration of 48 h in the presence or absence of 50 μg/mL of AXT. Subsequent to the incubation period, cells were harvested via trypsinization, with 1 × 10^6^ cells being transferred into E-tubes. Cold phosphate-buffered saline (PBS) was employed to wash the cells, followed by fixation in ice-cold 70% ethanol at −20 °C for a minimum of 3 h. Subsequently, 100 μL of the fixed cells was transferred into fresh E-tubes and washed with PBS. The cells were then resuspended in 200 μL of the Muse^TM^ Cell Cycle Kit (Millipore Co., Darmstadt, Germany) and incubated for 30 min in darkness. The resultant samples were subjected to analysis utilizing the Muse^TM^ Cell Analyzer.

### 3.6. Cell Apoptosis Analysis

Cell apoptosis was assessed using the MUSE^TM^ Cell Analyzer system. Cells were cultured for 48 h in the presence or absence of 50 μg/mL of AXT. Following incubation, cells were harvested via trypsinization, and 1 × 10^5^ cells were aliquoted into E-tubes. Subsequently, cells were washed with cold PBS supplemented with 1% FBS and resuspended in 100 μL of the washing solution. Next, 100 μL of the Muse^TM^ Annexin V εt Dead Cell Kit (Millipore Co.) was added, and the mixture was incubated for 20 min in the absence of light. Finally, the samples were analyzed using the Muse^TM^ Cell Analyzer.

### 3.7. Statistical Analysis

Statistical analysis (Kruskal–Wallis test, a non-parametric test) was performed using PRISM version 5.0 (GraphPad software, San Diego, CA, USA). The *p*-value was deemed significant if it was less than 0.05.

## 4. Discussion

Cancer poses a significant global health threat, impacting populations not only within Korea but worldwide, with diverse etiological factors implicated. Through extensive research efforts by numerous scientists, considerable insights into cancer biology have been gained. Various therapeutic modalities and pharmacological agents have been investigated, demonstrating a range of efficacies. Among these, AXT, a potent antioxidant extractable from marine organisms, has been reported to possess anti-tumor, antioxidative, anti-inflammatory, and neuroprotective properties. The association between astaxanthin and the pivotal tumor suppressor protein p53 has garnered increasing attention in cancer research. Our study findings demonstrate that AXT inhibits proliferation, induces cell cycle arrest, and promotes apoptosis in breast cancer cells. Furthermore, we hypothesize that these effects are mediated through the AXT-induced upregulation of p53. Prior studies have reported that AXT suppresses pontin and mutant p53 expression in BT20 and T47D breast cancer stem cells, thereby modulating cell cycle regulation [14]. Additionally, AXT has been shown to inhibit the expression of proteins antagonistic to p53, such as MDM2, thereby stabilizing and activating p53 in cancer cells [15]. Moreover, emerging evidence suggested that AXT could synergize with conventional chemotherapy agents to enhance efficacy through regulation of the p53-mediated pathway [1,9,15]. These studies are supported by the notion that AXT affects the cell cycle of MCF-7 breast cancer cells via the p53/p21 pathway, and induces apoptosis of the tumor cells via the activation of caspase cascades. We acknowledge that our study has a limitation due to its reliance on a single cell line, which may compromise its reliability. However, previous reports have conducted experiments using both non-tumorigenic and malignant cell lines [16]; we refer to these studies for further discussion. In the study, it was found that treatment with AXT induced apoptosis in malignant cell lines (T47D and MDA-MB-231), while having negligible effects on non-tumorigenic cell lines (MCF-10A). This supports our findings that the treatment influences the cell cycle, leading to DNA changes and errors, which subsequently trigger apoptosis.

In conclusion, our study elucidates that AXT induces p53 expression and caspase-3 activation in MCF-7 cells, with the resultant upregulation of p53 mediating cell cycle arrest and apoptosis. While further research is necessary to fully elucidate the molecular mechanisms underpinning the relationship between astaxanthin and p53, our current findings suggest that astaxanthin holds promise as a potential adjuvant therapy in cancer treatment, targeting the p53 signaling pathway.

## Figures and Tables

**Figure 1 ijms-25-07111-f001:**
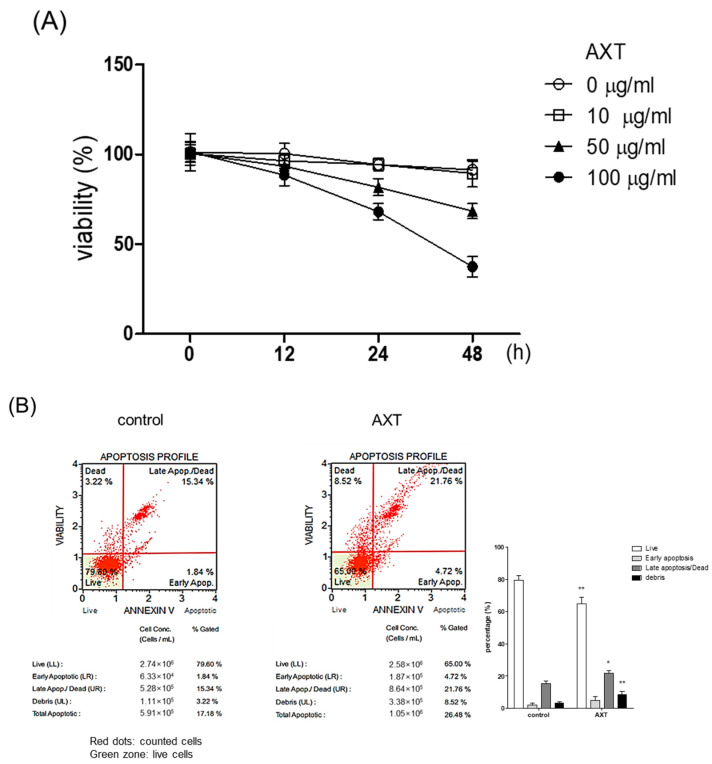
Effect of AXT on the viability of MCF-7 cell. (**A**) MCF-7 cells (5 × 10^3^ cells/well) were incubated with or without 10, 50, or 100 μg/mL of AXT for indicated periods in 96-well plate. After the incubation, PrestoBlue^TM^ was added in the wells, and was measured with a spectrophotometer at 540 nm wavelength. Data are expressed as means ± SD (*n* = 3 replicates/group). (**B**) The cells were incubated with 50 μg/mL of AXT for 48 h, and the cells were harvested. The processed samples according to the apoptosis analysis kit manual were analyzed by Muse^TM^ Cell Analyzer. The bar graph represents the results of the FACS analysis. Data are expressed as means ± SD (*n* = 3 replicates/group). (**: *p* < 0.01, *: *p* < 0.05 vs. same group in control). Results are representative of 3 independent experiments.

**Figure 2 ijms-25-07111-f002:**
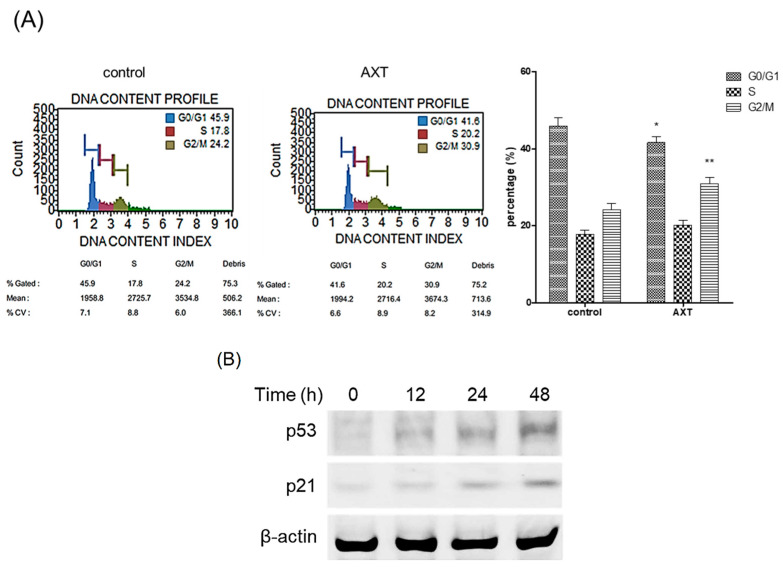
Analysis of DNA contents and p53/p21 expression by AXT. (**A**) MCF-7 cells were incubated with 50 μg/mL of AXT for 48 h, and the cells were harvested. The samples processed according to the DNA content analysis kit manual were analyzed with a Muse^TM^ Cell Analyzer. Results are representative of 3 independent experiments. The bar graph represents the results of the DNA contents analysis. Data are expressed as means ± SD (*n* = 3 replicates/group). (**: *p* < 0.01, *: *p* < 0.05 vs. the same group in the control). (**B**) MCF-7 cells were incubated with 50 μg/mL of AXT for 0, 12, 24, and 48 h, and protein were harvested from the cells. The total proteins were analyzed with Western blot. β-Actin used as loading control. Results are representative of 3 independent experiments.

**Figure 3 ijms-25-07111-f003:**
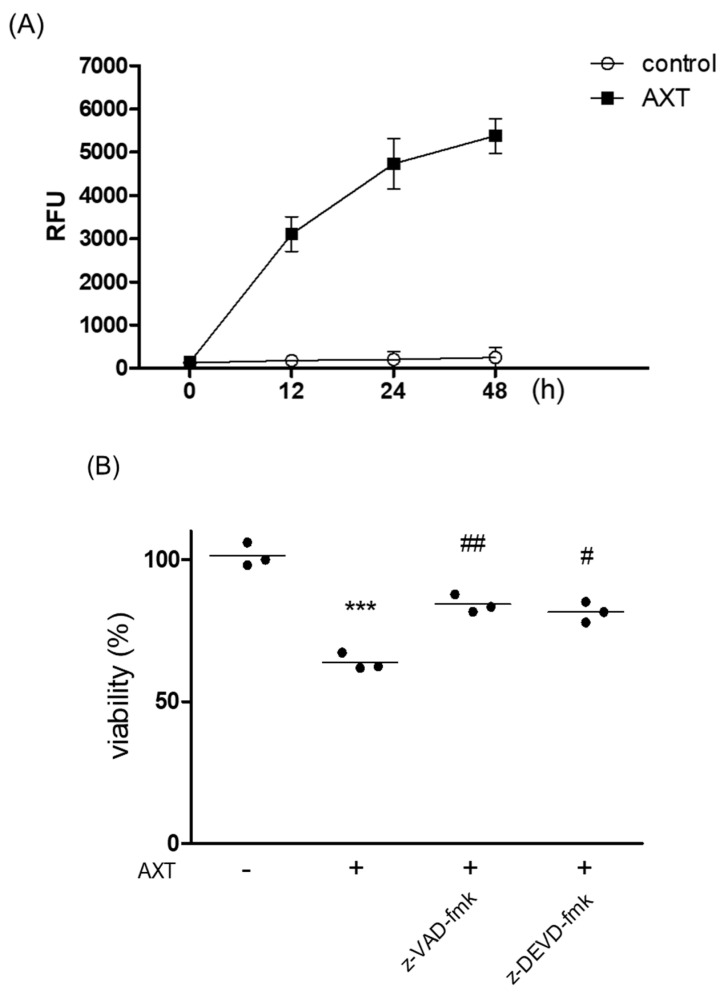
Analysis of the caspase involvement by AXT. (**A**) MCF-7 cells (5 × 10^4^ cells/well) in a 96-well plate were exposed to 50 μg/mL of AXT for indicated times, and the cells were lysed and assessed by following the user manual of the Caspase-3 Activity Assay Kit. The absorbances were measured at 380 nm. Data are expressed as means ± SD (*n* = 3 replicates/group). Results are representative of 3 independent experiments. (**B**) The cells (5 × 10^3^ cells/well) were pre-treated with 10 μM of z-VAD-fmk or z-DEVD-fmk for 2 h, and were incubated with 50 μg/mL of AXT for 48 h in 96-well plate. After the incubation, PrestoBlue^TM^ was added in the wells, and was measured with a spectrophotometer at 540 nm wavelength. Data are expressed as means ± SD (*n* = 3 replicates/group). (***: *p* < 0.005 vs. control; ##: *p* < 0.01, #: *p* < 0.05 vs. AXT).

## Data Availability

All data generated or analyzed during this study are included in this published article.

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
