# Peer review of "Astaxanthin Induces Apoptosis in MCF-7 Cells through a p53-Dependent Pathway"

_ijms, 2024, doi:10.3390/ijms25137111_

Round 1

Reviewer 1 Report

Comments and Suggestions for Authors

1- the study could be improved by invistgating more than one breast cancer cell lines indicating different stages of the disease  

2- for the viability assay, it would be helpful to find the IC50, this would help in using the correct concentration in the other expermints!

3- in apoptosis test, cell-cycle test: i do not think there were significant difference between the control and ATX, this should be stated clearly or imporve that there is statistic difference (P value!) 

Author Response

< Answers to Reviewer 1 >

  1. The study could be improved by investigating more than one breast cancer cell lines indicating different stages of the disease.

A: Thank you for highlighting this crucial point. We acknowledge that utilizing a single cell line represents a limitation of our study. However, due to current constraints, we are unable to perform additional experiments with other cell lines. Notably, previous research has already examined the effects of AXT on non-neoplastic cell lines (MCF-10A) as well as malignant tumor cell lines (T47D and MDA-MB-231). We will incorporate this information into the discussion section.

  1. For the viability assay, it would be helpful to find the IC50, this would help in using the correct concentration in the other experiments!

A: Thank you for your important advice. We add the IC50 data in supplementary data.

  1. 3. In apoptosis test, cell-cycle test: I do not think there were significant difference between the control and ATX, this should be stated clearly or improve that there is statistic difference (P value!)

A: Thank you for your important and pertinent advice. We have enhanced the presentation of the results by illustrating them with bar graphs for greater clarity. Additionally, we have provided the p-values. Thank you.

Reviewer 2 Report

Comments and Suggestions for Authors

This paper describes the study on a breast cancer treatment candidate, astaxanthin (AXT), that is a xanthophyll carotenoid with antioxidant properties. AXT is obtained from some marine organisms. Unfortunately, the Authors have performed the experiments on one cell line only, namely MCF-7, a human breast adenocarcinoma cells representing the luminal A breast cancer subtype that is characterized with lowest malignancy as compared with other breast cancer subtypes. Such a study should also be done on a non-cancerous mammary gland epithelial cell line (MCF-10A, MCF-12A, etc.) to estimate whether AXT induces the same pro-apoptotic mechanisms in normal mammary gland epithelial cells. Moreover, the same experiments should be performed on more malignant breast cancer cells, for example SK-BR-3 (a model of HER2-positive breast cancer) and/or MDA-MB-231 (a cell line representing triple-negative breast cancer). The Authors should perform the corresponding experiments on previously mentioned  cell lines (a non-tumorigenic cell line is the absolute minimum), submit the manuscript again and then the paper can be reconsidered. Moreover, I have found other weak points that should be corrected.

1. In “Materials and Methods” sub-section “Western blot” nothing is mentioned about the membrane stripping method for a loading control (beta-actin) detection. This information should be completed.

2. In “Materials and Methods” sub-section “Cell viability assay” on page 2, lines 82-83 the Authors wrote: “These cells were then exposed to varying concentrations of AXT in growth media as indicated…” – it is not specified, where this information is indicated.

3. In “Materials and Methods” sub-section “Statistical analysis”: what was the reason to use one-way ANOVA (parametric test) for the analysis? Was this decision supported by a normality test? If such a test was not performed, a non-parametric test (Kruskal-Wallis) should be used. Moreover, the Authors have not mentioned, what kind of post-hoc test was used to calculate the p-values and at what p-value the differences between the tested groups were considered to be statistically relevant?

4. The Authors have written, that they have estimated cell viability and proliferation by Presto Blue assay. Actually, this assay measures the cell viability only, for cell proliferation another technique should be used, for example BrdU incorporation assay. The authors should perform a cell proliferation assay or it should be stated in the manuscript, that only the cell viability was estimated.

5. Plot in Figure 3B: the bar charts are widely used, but this is inappropriate in terms of statistics. Instead, a dot plot or boxes with whiskers should be used – please change the graph type in Fig. 3B due to my suggestions.

Comments on the Quality of English Language

Even though the English language editing was done by a specialized service, the moderate English language corrections are still required:

- page 1, line 21: switch the tenses in the following fragment: change ‘…it is studied that AXT acted as…’ to ‘…it was studied that AXT acts as…’;

- page 1, line 25: change ‘…the tumor cells were induced the apoptosis…’ to ‘…the apoptosis was induced in tumor cells…’;

- page 1, line 30: change ‘And our results will provide…’ to ‘Thus, our results will provide…’;  

- page 2, line 64: change ‘z-VAD-fmk and z-DEVD-fmk was purchased…’ to ‘z-VAD-fmk and z-DEVD-fmk were purchased…’;

- page 3, lines 125-126: poor English style (repetition) in the following fragment: ‘…in order to ascertain the role of AXT in p53-dependent cellular apoptosis, we conducted an analysis of cellular apoptosis.’ – this fragment should be written in another way to avoid the repetition of ‘cellular apoptosis’ phrase ;

- page 7, line 195: poor English style (repetition) in the following fragment: ‘…cancer research. Our research…’, please change to: ‘…cancer research. Our study…’.

Author Response

< Answers to Reviewer 2 >

This paper describes the study on a breast cancer treatment candidate, astaxanthin (AXT), that is a xanthophyll carotenoid with antioxidant properties. AXT is obtained from some marine organisms. Unfortunately, the Authors have performed the experiments on one cell line only, namely MCF-7, a human breast adenocarcinoma cells representing the luminal A breast cancer subtype that is characterized with lowest malignancy as compared with other breast cancer subtypes. Such a study should also be done on a non-cancerous mammary gland epithelial cell line (MCF-10A, MCF-12A, etc.) to estimate whether AXT induces the same pro-apoptotic mechanisms in normal mammary gland epithelial cells. Moreover, the same experiments should be performed on more malignant breast cancer cells, for example SK-BR-3 (a model of HER2-positive breast cancer) and/or MDA-MB-231 (a cell line representing triple-negative breast cancer). The Authors should perform the corresponding experiments on previously mentioned cell lines (a non-tumorigenic cell line is the absolute minimum), submit the manuscript again and then the paper can be reconsidered. Moreover, I have found other weak points that should be corrected.

A: Thank you for highlighting this crucial point. We acknowledge that utilizing a single cell line represents a limitation of our study. However, due to current constraints, we are unable to perform additional experiments with other cell lines. Notably, previous research has already examined the effects of AXT on non-neoplastic cell lines (MCF-10A) as well as malignant tumor cell lines (T47D and MDA-MB-231). We will incorporate this information into the discussion section.

  1. In “Materials and Methods” sub-section “Western blot” nothing is mentioned about the membrane stripping method for a loading control (beta-actin) detection. This information should be completed.

A: Thank you for your advice. We add the mentions.

  1. In “Materials and Methods” sub-section “Cell viability assay” on page 2, lines 82-83 the Authors wrote: “These cells were then exposed to varying concentrations of AXT in growth media as indicated…” – it is not specified, where this information is indicated.

A: Yes, we correct the sentence.

  1. In “Materials and Methods” sub-section “Statistical analysis”: what was the reason to use one-way ANOVA (parametric test) for the analysis? Was this decision supported by a normality test? If such a test was not performed, a non-parametric test (Kruskal-Wallis) should be used. Moreover, the Authors have not mentioned, what kind of post-hoc test was used to calculate the p-values and at what p-value the differences between the tested groups were considered to be statistically relevant?

A: Thank you for your advice. We use the Prism program to generate graphs and statistical results. As we are not experts in statistics, we are not fully aware of the differences between various methods. We will proceed with statistical processing in the manner you mentioned. Leveraging your high-quality advice, we will conduct more thorough statistical research in our next study and strive to produce more robust results. We have also included sentences pertaining to p-values and significance levels.

  1. The Authors have written, that they have estimated cell viability and proliferation by Presto Blue assay. Actually, this assay measures the cell viability only, for cell proliferation another technique should be used, for example BrdU incorporation assay. The authors should perform a cell proliferation assay or it should be stated in the manuscript, that only the cell viability was estimated.

A: Your mention is correct. We fix the part.

  1. Plot in Figure 3B: the bar charts are widely used, but this is inappropriate in terms of statistics. Instead, a dot plot or boxes with whiskers should be used – please change the graph type in Fig. 3B due to my suggestions.

A: Yes, we changed the graph according to your suggestion.

Comments on the Quality of English Language

Even though the English language editing was done by a specialized service, the moderate English language corrections are still required:

A: Yes, we correct the sentences according your advices. Thank you very much.

- page 1, line 21: switch the tenses in the following fragment: change ‘…it is studied that AXT acted as…’ to ‘…it was studied that AXT acts as…’;

A: Done.

- page 1, line 25: change ‘…the tumor cells were induced the apoptosis…’ to ‘…the apoptosis was induced in tumor cells…’;

A: Done.

- page 1, line 30: change ‘And our results will provide…’ to ‘Thus, our results will provide…’; 

A: Done.

- page 2, line 64: change ‘z-VAD-fmk and z-DEVD-fmk was purchased…’ to ‘z-VAD-fmk and z-DEVD-fmk were purchased…’;

A: Done.

- page 3, lines 125-126: poor English style (repetition) in the following fragment: ‘…in order to ascertain the role of AXT in p53-dependent cellular apoptosis, we conducted an analysis of cellular apoptosis.’ – this fragment should be written in another way to avoid the repetition of ‘cellular apoptosis’ phrase ;

A: Done.

- page 7, line 195: poor English style (repetition) in the following fragment: ‘…cancer research. Our research…’, please change to: ‘…cancer research. Our study…’.

A: Done.

Round 2

Reviewer 1 Report

Comments and Suggestions for Authors

previous comments was not addressed properly, 

There is a problem in the design of the method:

IC50 cannot be calculated from just 3 points

The reason for using the concentrations in the experiments has not been explained

The p-Value calculation is not resonable 

Author Response

  1. previous comments were not addressed properly,

There is a problem in the design of the method:

IC50 cannot be calculated from just 3 points

A: Thank you for your insightful feedback. Our hasty approach led to non-optimal results. We have since conducted additional experiments with increased concentrations and have determined the IC50. This information adds to the main text. Thank you.

The reason for using the concentrations in the experiments has not been explained

A: We conducted experiments using various concentrations of AXT and presented the results. From these, we selected the concentration at which the survival rate of the cancer cells was approximately 70% for subsequent experiments. This information adds to the main text. Thank you.

The p-Value calculation is not resonable

A: Thank you for your important advice. We corrected the wrong expressions. Thank you.

Reviewer 2 Report

Comments and Suggestions for Authors

The Authors of the manuscript have made the corrections according to my suggestions and the paper is now ready for publication without any further changes. Thank you.

Author Response

The Authors of the manuscript have made the corrections according to my suggestions and the paper is now ready for publication without any further changes. Thank you.

A: Your advice has enabled me to write a higher-quality paper. Thank you for your valuable guidance.

Round 3

Reviewer 1 Report

Comments and Suggestions for Authors

1- The IC50 dose not include SEM!

2- the graphs resolution (fig1-B, Fig 2-A&B,  is low!

Author Response

1- The IC50 dose not include SEM!

A: We apologize for our mistake. Following your advice, we have removed it. Thank you.

2- the graphs resolution (fig1-B, Fig 2-A&B,  is low!

A: The clarity and contrast of the figures have been adjusted. Thank you for the excellent advice.
